# Meta-Learning Sparse Compression Networks

**Jonathan Richard Schwarz**                                     *schwarzjn@google.com*
*DeepMind*
*University College London*

**Yee Whye Teh**                                                 *ywteh@google.com*
*DeepMind*

**Reviewed on OpenReview:** *https://openreview.net/forum?id=Cct7kqbHK6*

## Abstract

Recent work in Deep Learning has re-imagined the representation of data as functions mapping from a coordinate space to an underlying continuous signal. When such functions are approximated by neural networks this introduces a compelling alternative to the more common multi-dimensional array representation. Recent work on such *Implicit Neural Representations* (INRs) has shown that - following careful architecture search - INRs can outperform established compression methods such as JPEG (e.g. Dupont et al., 2021). In this paper, we propose crucial steps towards making such ideas scalable: Firstly, we employ state-of-the-art network sparsification techniques to drastically improve compression. Secondly, introduce the first method allowing for sparsification to be employed in the inner-loop of commonly used Meta-Learning algorithms, drastically improving both compression and the computational cost of learning INRs. The generality of this formalism allows us to present results on diverse data modalities such as images, manifolds, signed distance functions, 3D shapes and scenes, several of which establish new state-of-the-art results.

## 1 Introduction

An emerging sub-field of Deep Learning has started to re-imagine the representation of data items: While traditionally, we might represent an image or 3D shape as a multi-dimensional array, continuous representations of such data appear to be a more natural choice for the underlying signal. This can be achieved by defining a functional representation: mapping from spatial coordinates (x, y) to (r, g, b) values in the case of an image. The problem of learning such a function is then simply a supervised learning task, for which we may employ a neural network - an idea referred to as Implicit Neural Representations (INRs). An advantage of this strategy is that the algorithms for INRs are data agnostic - we may simply re-define the coordinate system and target signal values for other modalities and readily apply the same procedure. Moreover, the learned function can be queried at any point, allowing for the signal to be represented at higher resolutions once trained. Finally, the size of the network representation can be chosen by an expert or as we propose in this work, an algorithmic method to be lower than the native dimensionality of the array representation. Thus, this perspective provides a compelling new avenue into the fundamental problem of data compression. INRs are particularly attractive in cases where array representations scale poorly with the discretisation level (e.g. 3D shapes) or the underlying signal is inherently continuous such as in neural radiance fields (NerF) (Mildenhall et al., 2020) or when discretisation is non-trivial, for example when data lies on a manifold.

So far, the difficulty of adopting INRs as a compression strategy has been a trade-off between network size and approximation quality requiring architecture search (e.g. Dupont et al., 2021) or strong inductive biases (e.g. Chan et al., 2021; Mehta et al., 2021). Furthermore, the cost of fitting a network to a single data point vastly exceeds the computational cost of standard compression methods such as JPEG (Wallace, 1992), an issue that is compounded when additional architecture search is required.

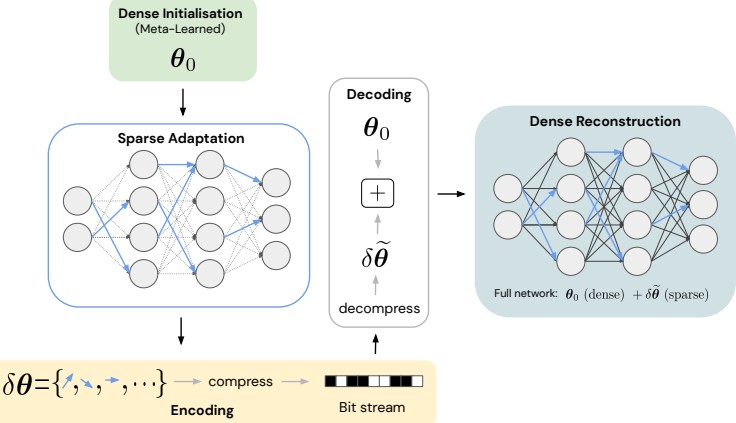

Figure 1: Overview of MSCN as a compression method. In order to compress a data item, we perform sparse adaptation of a meta-learned initialisation, leading to a small subset of changes $\delta\boldsymbol{\theta}$ encoding the item. Sparsity reduces compression cost and avoids costly architecture search. Meta-Learning drastically cuts compression time. $\delta\boldsymbol{\theta}$ can be subsequently compressed and encoded using standard techniques.

In this work, we specifically focus on improving the suitability of INRs as a compression method by tackling the aforementioned problems. First, we recognise insights of recent deep learning studies (e.g. Frankle & Carbin, 2018) which show how only a small subset of parameters encode the predictive function. We thus employ recent state-of-the-art sparsity techniques (Louizos et al., 2017) to explicitly optimise INRs using as few parameters as possible, drastically improving their compression cost.

Secondly, in order to tackle the computational cost of learning INRs, we follow recent work (e.g. Lee et al., 2021; Tancik et al., 2021; Dupont et al., 2022a) by adopting Meta-Learning techniques such as MAML (Finn et al., 2017) which allow learning an INR representing a single signal by fine-tuning from a learned initialisation using only a handful of optimisation steps. Crucially, our sparsity procedure allow efficient Backpropagation through this learning procedure, resulting in an initialisation that is specifically optimised for sparse signals. This allows us to re-imagine the sparsity procedure as uncovering a network structure most suitable for the task at hand. This is noticeably different from related work which decouples Meta-Learning from the sparsity procedure (Tian et al., 2020; Lee et al., 2021).

Figure 1 shows an overview of our technique for compression. The key novelty of our work lies in the sparse adaptation phase, significantly reducing compression cost.

Finally, our framework is flexible and allows for Weight-, Representation-, Group- and Gradient-Sparsity with minimal changes and is thus suitable for many applications outside of the core focus of this work.

## 2 Background

We start by reviewing relevant aspects of the Meta-Learning and INR literature that will constitute the fundamental building blocks of our method.

### 2.1 Implicit Neural Representations

Throughout the manuscript, we represent INRs as functions $f_{\boldsymbol{\theta}} : \mathcal{X} \to \mathcal{Y}$ parameterised by a neural network with parameters $\boldsymbol{\theta}$ mapping from a coordinate space $\mathcal{X}$ to the output space $\mathcal{Y}$. INRs are instance-specific, i.e. they are unique networks representing single data items. Its approximation quality is thus measured across all coordinates making up the data item, represented by the discrete index set $\mathcal{I}$. As an example, for a 3D shape the coordinate space is $\mathcal{X} = \mathbb{R}^3$ (x, y, z) and $\mathcal{Y} = [0, 1]$ are Voxel occupancies. $\mathcal{I}$ the 3D-grid $\{0, \ldots, D\}^3$ where $D$ is the discretisation level. We can thus formulate the learning of an INR as a minimisation problem of the squared error between the INR's prediction and the underlying signal:

$$\min_{\boldsymbol{\theta}} \mathcal{L}(f_{\boldsymbol{\theta}}, \mathbf{x} \in \mathcal{X}, \mathbf{y} \in \mathcal{Y}) = \min_{\boldsymbol{\theta}} \sum_{i \in \mathcal{I}} ||f_{\boldsymbol{\theta}}(\mathbf{x}_i) - \mathbf{y}_i||_2^2 \qquad (1)$$

which is typically minimised via Gradient Descent. As a concrete choice for $f_{\boldsymbol{\theta}}$, recent INR breakthroughs propose the combination of Multi-Layer Perceptrons (MLPs) with either positional encodings (Mildenhall et al., 2020; Tancik et al., 2020) or sinusoidal activation functions (Sitzmann et al., 2020b). Both methods significantly improve on the reconstruction error of standard ReLU Networks (Nair & Hinton, 2010).

Of particular importance to the remainder of our discussion around compression is the recent observation that signals can be accurately learned using merely data-item specific modulations to a shared base network (Perez et al., 2018; Mehta et al., 2021; Dupont et al., 2022a). Specifically, in the forward pass of a network, each layer $l$ represents the transformation $\mathbf{x} \mapsto f(\mathbf{W}^{(l)}\mathbf{x} + \mathbf{b}^{(l)} + \mathbf{m}^{(l)})$, where $\{\mathbf{W}^{(l)}, \mathbf{b}^{(l)}\}$ are weights and biases shared between signal with only modulations $\mathbf{m}^{(1)}$ being specific to each signal. This has the advantage of drastically reducing compression costs and is thus of particular interest to the problem considered by us.

## 2.2   Meta-Learning

It is worth pointing out that the minimisation of Equation (1) is extraordinarily expensive: Learning a single NeRF ((Mildenhall et al., 2020)) scene can take up to an entire day on a single GPU (Dupont et al., 2022a); even the compression of a single low-dimensional image requires thousands of iterative optimisation steps. Fortunately, we need not resort to Tabula rasa optimisation for each data item in turn: In recent years, developments in Meta-Learning (e.g. Thrun & Pratt, 2012; Andrychowicz et al., 2016) have provided a formalism that allow a great deal of learning to be shared among related tasks or in our case signals in a dataset.

Model-agnostic Meta Learning (MAML) (Finn et al., 2017) provides an optimisation-based approach to finding an initialisation from which we can specialise the network to an INR representing a signal in merely a handful of optimisation steps. Following custom notation, we will consider a set of tasks or signals $\{\mathcal{T}_1, \ldots, \mathcal{T}_n\}$. Finding a minimum of the loss on each signal is achieved by Gradient-based learning on the task-specific data $(\mathbf{x}_{\mathcal{T}_i}, \mathbf{y}_{\mathcal{T}_i})$. Writing $\mathcal{L}_{\mathcal{T}_i}(f_{\boldsymbol{\theta}})$ as a shorthand for $\mathcal{L}(f_{\boldsymbol{\theta}}, \mathbf{x}_{\mathcal{T}_i}, \mathbf{y}_{\mathcal{T}_i})$, a single gradient step from a shared initialisation $\boldsymbol{\theta}_0$ takes the form:

$$\boldsymbol{\theta}_i' = \boldsymbol{\theta}_0 - \beta \nabla_{\boldsymbol{\theta}} \mathcal{L}_{\mathcal{T}_i}(f_{\boldsymbol{\theta}}) \qquad (2)$$

which we can trivially iterate for multiple steps. The key insight in MAML is to define a Meta-objective for *learning* the initialisation $\boldsymbol{\theta}_0$ as the minimisation of an expectation of the task-specific loss after its update:

$$\boldsymbol{\theta}_0 = \arg\min_{\boldsymbol{\theta}} \mathbb{E}_{\mathcal{T}_i \sim p(\mathcal{T})} \mathcal{L}_{\mathcal{T}_i}(f_{\boldsymbol{\theta}_i'}) = \arg\min_{\boldsymbol{\theta}} \mathbb{E}_{\mathcal{T}_i \sim p(\mathcal{T})} \mathcal{L}_{\mathcal{T}_i}(f_{\boldsymbol{\theta} - \beta \nabla_{\boldsymbol{\theta}} \mathcal{L}_{\mathcal{T}_i}(f_{\boldsymbol{\theta}})}) \qquad (3)$$

where $p(\mathcal{T})$ is a distribution over signals in a dataset. The iterative optimisation of (3) is often referred to the MAML "outer loop" and (2) as the "inner loop" respectively. Note that this is a second-order optimisation objective requiring the differentiation through a learning process, although first-order approximations exist (Nichol & Schulman, 2018).

Indeed, this procedure has been widely popular in the work on INRs, e.g. being successfully used for NeRF scenes in Tancik et al. (2021) or signed distance functions (Sitzmann et al., 2020a). Finally, it should be noted that the idea of learning modulations explored in the previous section relies on the MAML process. Thus, the learning of weights and biases is achieved using (3), although only modulations are adopted in the inner-loop (2). Meta-learning a subset of parameters in the inner-loop corresponds to the MAML-derivative CAVIA (Zintgraf et al., 2019).

## 3 Meta-Learning Sparse Compression Networks

This section introduces the key contributions of this work, providing a framework for sparse Meta-Learning which we instantiate in two concrete algorithms for learning INRs.

### 3.1 $L_0$ Regularisation

While the introduction of sparsity in the INR setting is highly attractive from a compression perspective, our primary difficulty in doing so is finding a procedure compatible with the MAML process described in the previous section. This requires (i) differentiability and (ii) fast learning of both parameters and the subnetwork structure.

We can tackle (i) by introducing $L_0$ Regularisation (Louizos et al., 2017), a re-parameterised $L_0$ objective using stochastic gates on parameters. Consider again the INR objective (1) with a sparse reparameterisation $\widetilde{\boldsymbol{\theta}}$ of a dense set of parameters $\boldsymbol{\theta}$: $\widetilde{\boldsymbol{\theta}} = \boldsymbol{\theta} \odot \mathbf{z}; z_j \in \{0, 1\}$ and an L0 Regularisation term on the gates with regularisation coefficient $\lambda$. We can learn a subnetwork structure by optimising distributional parameters $\boldsymbol{\pi}$ of a distribution $q(\mathbf{z}|\boldsymbol{\pi})$ on the gates leading to the regularised objective:

$$\min_{\boldsymbol{\theta}, \boldsymbol{\pi}} \mathcal{L}^{\mathcal{R}}(f_{\widetilde{\boldsymbol{\theta}}}, \mathbf{x}, \mathbf{y}, \boldsymbol{\pi}) = \mathbb{E}_{q(\mathbf{z}|\boldsymbol{\pi})} \Big[ \sum_{i \in \mathcal{I}} ||f_{\boldsymbol{\theta} \odot \mathbf{z}}(\mathbf{x}_i) - \mathbf{y}_i||_2^2 \Big] + \lambda \sum_{j=1}^{\dim(\boldsymbol{\pi})} \boldsymbol{\pi}_j \tag{4}$$

where $\lambda \sum_{j=1}^{\dim(\boldsymbol{\pi})} \boldsymbol{\pi}_j$ penalises the probability of gates being non-zero.

The key on overcoming non-differentiability due to the discrete nature of $\mathbf{z}$ is smoothing the objective: This is achieved by choosing an underlying continuous distribution $q(\mathbf{s}|\boldsymbol{\phi})$ and transforming its random variables by a hard rectification: $\mathbf{z} = \min(1, \max(0, \mathbf{s})) = g(\mathbf{s})$. In addition, we note that the L0 penalty can be naturally expressed by using the CDF of $q$ to penalise the probability of the gate being non zero $(1 - Q(\mathbf{s} \leq 0|\boldsymbol{\phi}_j))$:

$$\min_{\boldsymbol{\theta}, \boldsymbol{\phi}} \mathcal{L}^{\mathcal{R}}(f_{\widetilde{\boldsymbol{\theta}}}, \mathbf{x}, \mathbf{y}, \boldsymbol{\phi}) = \mathbb{E}_{q(\mathbf{s}|\boldsymbol{\phi})} \Big[ \sum_{i \in \mathcal{I}} ||f_{\boldsymbol{\theta} \odot g(\mathbf{s})}(\mathbf{x}_i) - \mathbf{y}_i||_2^2 \Big] + \lambda \sum_{j=1}^{\dim(\boldsymbol{\phi})} 1 - Q(\mathbf{s}_j \leq 0|\boldsymbol{\phi}_j) \tag{5}$$

A suitable choice for $q(\mathbf{z}|\boldsymbol{\pi})$ is a distribution allowing for the reparameterisation trick (Kingma & Welling, 2013), i.e. the expression of the expectation in (5) as an expectation of a parameter-free noise distribution $p(\epsilon)$ from which we obtain samples of $s$ through a transformation $f(\cdot; \phi)$. This allows a simple Monte-Carlo estimation of (5):

$$\min_{\boldsymbol{\theta}, \boldsymbol{\phi}} \mathcal{L}^{\mathcal{R}}(f_{\widetilde{\boldsymbol{\theta}}}, \mathbf{x}, \mathbf{y}, \boldsymbol{\phi}) = \frac{1}{S} \sum_{i=1}^{S} \Big[ \sum_{i \in \mathcal{I}} ||f_{\boldsymbol{\theta} \odot g(f(\boldsymbol{\epsilon}_s, \boldsymbol{\phi}))}(\mathbf{x}_i) - \mathbf{y}_i||_2^2 \Big] + \lambda \sum_{j=1}^{\dim(\boldsymbol{\phi})} 1 - Q(\mathbf{s}_j \leq 0|\boldsymbol{\phi}_j); \epsilon_s \sim p(\boldsymbol{\epsilon}) \tag{6}$$

The choice for $q(\mathbf{s}|\boldsymbol{\phi})$ in Louizos et al. (2017) is the Hard concrete distribution, obtained by stretching the concrete (Maddison et al., 2016) allowing for reparameterisation, evaluation of the CDF and exact zeros in the gates/masks $\mathbf{z}$. A suitable estimator is chosen at test time.

### 3.2 Sparsity in the inner loop

We are now in place to build our method from the aforementioned building blocks. Concretely, consider the application of $L_0$ Regularisation in the MAML inner loop objective. Using a single Monte-Carlo sample for simplicity, can re-write the MAML meta-objective as:

$$\boldsymbol{\theta}_0 = \arg\min_{\boldsymbol{\theta}} \mathbb{E}_{\mathcal{T}_i \sim p(\mathcal{T})} \Big[ \mathcal{L}_{\mathcal{T}_i}^{\mathcal{R}}(f_{\boldsymbol{\theta}', \boldsymbol{\phi}'}) \Big] = \arg\min_{\boldsymbol{\theta}} \mathbb{E}_{\mathcal{T}_i \sim p(\mathcal{T})} \Big[ \mathcal{L}_{\mathcal{T}_i}(f_{(\boldsymbol{\theta} + \delta\boldsymbol{\theta}) \odot \mathbf{z}'}) + \lambda \sum_{j=1}^{\dim(\boldsymbol{\phi})} 1 - Q(\mathbf{s}_j \leq 0 | \boldsymbol{\phi}'_j) \Big] \quad (7)$$

$$\mathbf{z}' = g(f(\boldsymbol{\epsilon}, \boldsymbol{\phi}')); \boldsymbol{\phi}' = \boldsymbol{\phi}_0 + \delta\boldsymbol{\phi} \text{ and } \boldsymbol{\epsilon} \sim p(\boldsymbol{\epsilon})$$

where $\delta\boldsymbol{\theta}$ and $\delta\boldsymbol{\phi}$ are updates to both parameters and gate distributions computed in the inner loop (2). Note that this constitutes a fully differentiable sparse Meta-Learning algorithm.

However, a moment of reflection reveals concerns with (7): (i) The joint learning of both model and mask parameters in the inner-loop is expected to pose a more difficult optimisation problem due to trade-off between regularisation cost and task performance. This is particularly troublesome as the extension of MAML to long inner-loops is computationally very expensive and hence still an active area of research (e.g. Flennerhag et al., 2019; 2021). (ii) The sparsification $(\boldsymbol{\theta} + \delta\boldsymbol{\theta}) \odot \mathbf{z}'$ is sub-optimal from a compression standpoint: As the signal-specific compression cost is $\delta\boldsymbol{\theta}$, there is no need to compute the INR using a sparse network, provided $\delta\boldsymbol{\theta}$ is sparse. $\theta_0$ is signal-independent set of parameters and its compression cost thus amortised.

Our remedy to (i) is take inspiration from improvements on MAML that propose learning further parameters of the inner optimisation process (Li et al., 2017) such as the step size (known as MetaSGD). In particular, we learn both an initial set of parameters $\boldsymbol{\theta}_0$ and gates $\boldsymbol{\phi}_0$ through the outer loop, i.e.

$$\boldsymbol{\theta}_0, \boldsymbol{\phi}_0 = \arg\min_{\boldsymbol{\theta}, \boldsymbol{\phi}} \mathbb{E}_{\mathcal{T}_i \sim p(\mathcal{T})} \Big[ \mathcal{L}_{\mathcal{T}_i}(f_{(\boldsymbol{\theta} + \delta\boldsymbol{\theta}) \odot g(\boldsymbol{\epsilon}, \boldsymbol{\phi} + \delta\boldsymbol{\phi})}) + \lambda \sum_{j=1}^{\dim(\boldsymbol{\phi})} 1 - Q(\mathbf{s}_j \leq 0 | (\boldsymbol{\phi} + \delta\boldsymbol{\phi})_j)) \Big] \quad (8)$$

This has interesting consequences: While we previously relied solely on the inner-loop adaptation procedure to "pick out" an appropriate network for each signal, we have now in effect reserved a sub-network that provides a particularly suitable initialisation for the set of signals at hand. This sub-network can either be taken to be fixed (such as when the adopted networks are used to learn a prior or generative model (Dupont et al., 2022a)) or adopted within an acceptable budget of inner steps, providing a mostly overlapping but yet signal-specific set of gates.

With regards to concern (ii) it is worth noting that gates $\mathbf{z}$ may be applied *at any point* in the network. This is attractive as it provides us with a simple means to implement various forms of commonly encountered sparsity: 1. Unstructured sparsity by a direct application to all parameters 2. Structured sparsity by restricting the sparsity pattern 3. Group sparsity by sharing a single gate among sets of parameters 4. representational sparsity by gating activations or 5. Gradient Sparsity by masking updates in the inner loop. This highlights a strength of our method: the principles discussed so far allow for a framework in which sparsity can be employed in a Meta-Learning process in a variety of different ways depending on the requirements of the problem at hand. We refer to this framework with a shorthand of this manuscript's title: MSCN.

With regards to compression considerations, a more natural objective would thus involve a term $(\boldsymbol{\theta}_0 + \delta\boldsymbol{\theta} \odot \mathbf{z})$ in the inner loop, which ensures that we directly optimise for per-signal performance *using an update to as few parameters as possible* - the real per-signal compression cost. Note that as $\boldsymbol{\theta}_0$ is dense, the resulting $\theta_0 + \delta\theta$ (sparse) is still dense, thus allowing the use of more capacity in comparison to a fully sparse network. We suggest two concrete forms of $\delta\boldsymbol{\theta}$ in the next section.

### 3.3 Forms of $\delta\theta$

#### 3.3.1 Unstructured Sparse Gradients

Perhaps the most natural form of implementing a direct sparsification of $\delta\boldsymbol{\theta}$ is through gating of the gradients. Assuming a budget of inner-loop steps $T$ and writing $\mathcal{L}^{\mathcal{R}}$ for the regularised $L_0$ objective and $\boldsymbol{\theta}_t$ for the state of the parameters after $t-1$ updates, this takes the form of unstructured sparse gradients:

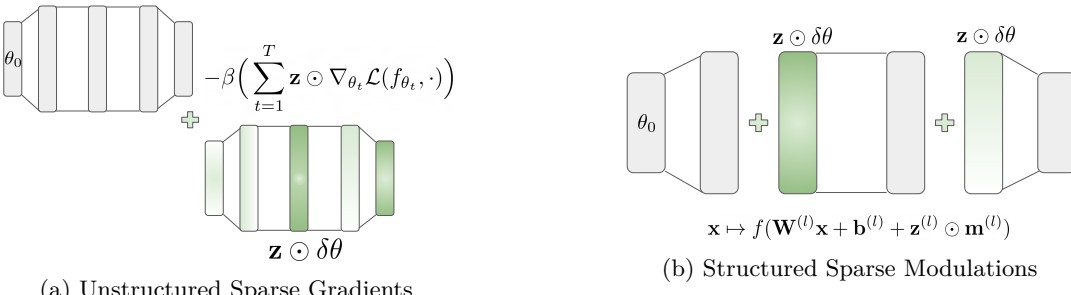

(a) Unstructured Sparse Gradients

(b) Structured Sparse Modulations

Figure 2: Instantiations of the MSCN framework.

$$\delta\boldsymbol{\theta} = -\beta \sum_{i=1}^{T} \mathbf{z} \odot \nabla_{\boldsymbol{\theta}_t} \mathcal{L}^{\mathcal{R}}(f_{\boldsymbol{\theta}_t}, \cdot) \tag{9}$$

and thus forces updates to concentrate on parameters with the highest plasticity in a few-shot learning setting. We impose no restrictions on the gates, applying them to both weight and biases gradients and thus allow for complex gradient sparsity patterns to emerge. In this setting, we found the use of aforementioned MetaSGD (Li et al., 2017) to be particularly effective.

### 3.3.2  Structured Sparse Modulations

An alternative form allows known inductive biases to inform our application of sparsity: In the second proposed instantiation of the MSCN framework we work in the case where only modulations $\{\mathbf{m}^{(l)}\}_{l=1}^{L}$ are allowed to adapt to each task-specific instance (see Section 2.1). The sparsification of those modulations is straight-forward and is achieved by introducing a layer-specific gate:

$$\mathbf{x} \mapsto f(\mathbf{W}^{(l)}\mathbf{x} + \mathbf{b}^{(l)} + \mathbf{z}^{(l)} \odot \mathbf{m}^{(l)}) \tag{10}$$

such that the only non-zero entries in $\delta\boldsymbol{\theta}$ are for sparse modulations. For ease of notation we omit zero-entries and simply write $\delta\boldsymbol{\theta} = \{\mathbf{z}^{(1)} \odot \mathbf{m}^{(1)}, \ldots, \mathbf{z}^{(L)} \odot \mathbf{m}^{(L)}\}$. This provides a particularly attractive form for compression due to the comparably low dimensionality of $\mathbf{m}^{(l)}$. Further sparsifying the modulations has the advantage of allowing the use of very deep or wide base networks $\boldsymbol{\theta}_0$, ensuring that common structure is modelled as accurately as possible while a small communication cost continues to be paid for $\delta\boldsymbol{\theta}$. Note also that an intuitive argument for deep networks is that modulations in early layers have a large impact on the rest of of the network. Both forms are shown in Figure 2.

## 4  Related work

### 4.1  Neural network sparsity

While dating back at least to the early 1990s, recent interest in sparsifying Deep Neural Networks has come both from experimental observations such as the lottery ticket hypothesis (Frankle & Carbin, 2018) and the unparalleled growth of models (e.g. Brown et al., 2020; Rae et al., 2021), often making the cost of training and inference prohibitive for all but the largest institutions in machine learning.

While more advanced methods have been studied (e.g. LeCun et al., 1989; Thimm & Fiesler, 1995), most contemporary techniques rely on the simple yet powerful approach of magnitude-based pruning - removing a pre-determined fraction by absolute value. Current techniques can be mainly categorised as the iterative sparsification of a densely initialised network (Gale et al., 2019) or techniques that maintain constant sparsity

throughout learning (e.g. Evci et al., 2020; Jayakumar et al., 2020). Critically, many recent techniques would not be suitable for learning in the inner-loop of a meta-learning algorithm due to non-differentiability, motivating our choice of a relaxed $L_0$ regularisation objective.

### 4.2 Sparse Meta-Learning

Despite Meta-Learning and Sparsity being well established research areas, the intersection of both topics has only recently started to attract increased attention. Noteworthy early works are Liu et al. (2019), who design a network capable of producing parameters for any sparse network structure, although an additional evolutionary procedure is required to determine well-performing networks. Also using MAML as a Meta-learner, Tian et al. (2020) employ weight sparsity as a means to improve generalisation.

Of particular relevance to this work is MetaSparseINR (Lee et al., 2021), who provide the first sparse Meta-Learning approach specifically designed for INRs. We can think of their procedure as the aforementioned iterative magnitude-based pruning technique (Ström, 1997; Gale et al., 2019) applied in the outer loop of MAML training. While has the advantage of avoiding the difficulty of computing gradients through a pruning operation, iterative pruning is known to produce inferior results (Schwarz et al., 2021) and limits the application to an identical sparsity pattern for each signal. Due to the direct relevance to our work, we will re-visit MetaSparseINR as a key baseline in the experimental Section.

### 4.3 Implicit Neural Representations

The compression perspective of INRs has recently been explored in the aforementioned COIN (Dupont et al., 2021) and in Davies et al. (2020) who focus on 3D shapes. Work on videos has naturally received increased attention, with (Chen et al., 2021) focusing on convolutional networks for video prediction (where the time stamp is provided as additional input) and Zhang et al. (2021) proposing to learn differences between frames via flow warping. While still lacking behind standard video codecs, fast progress is being made and early results are encouraging. An advantage of our contribution is that the suggested procedures can be readily applied to almost all deep-learning based architectures.

The recently proposed Functa (Dupont et al., 2022a) is a key baseline in our work as it also works on the insights of the modulation-based approach. Rather than using sparsity, the authors introduce a second network which maps a low dimensional latent vector to the full set of modulations. The instance-specific communication cost is thus the dimensionality of that latent vector. A minor disadvantage is the additional processing cost of running the latent vector to modulations network.

A particularly interesting observation is that quantisation of such modulations is highly effective (Strümpler et al., 2021; Dupont et al., 2022b), showing that simple uniform quantisation and arithmetic coding can significantly improve compression results.

Finally, it is also worth noting that work on INRs is related to the literature on multimodal architectures which have so far been mostly implemented through modality-specific feature extractors (e.g Kaiser et al., 2017), although recent work has used a single shared architecture (Jaegle et al., 2021).

## 5 Experiments

We now provide an extensive empirical analysis of the MSCN framework in the two aforementioned instantiations. Our analysis covers a wide range of datasets and modalities ranging from images to manifolds, Voxels, signed distance functions and scenes. While network depth and width vary based on established best practices, in all cases we use SIREN-style (Sitzmann et al., 2020b) sinusoidal activations. Throughout the section, we make frequent use of the Peak signal-to-noise ratio (PSNR), a metric commonly used to quantify reconstruction quality. For data standardised to the $[0, 1]$ range this is defined as: $\text{PSNR} = -\log_{10}(\text{MSE})$ where MSE is the mean-squared error. Further experimental details such as data processing steps and hyperparameter configurations in the Appendix.

## 5.1 Unstructured sparse Gradients

In the unstructured sparsity case (Section 3.3.1), the closest related work is MetaSparseINR (Lee et al., 2021) which will be the basis for our evaluations. Experiments in this section focus on Images, covering the CelebA (Liu et al., 2015) & ImageNette (Howard, 2022) datasets as well as Signed Distance Functions (Tancik et al., 2021), all widely used in the INR community. All datasets have been pre-processed to a size of $178 \times 178$. We follow the MetaSparseINR authors in the choice of those datasets and thus compare directly to that work as well as the baselines discussed. We provide a description of those baselines in the Appendix.

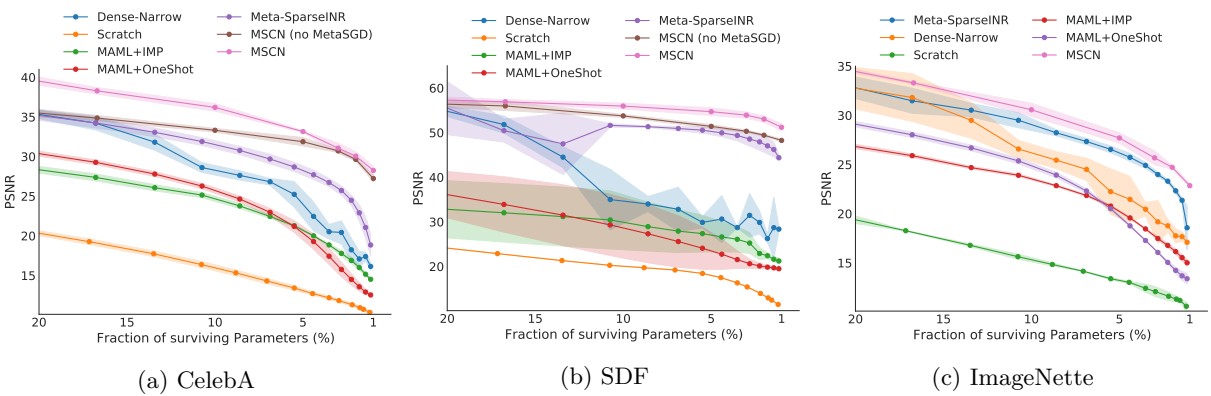

Figure 3: Quantitative results for unstructured sparsity. Results show the mean and standard deviation over three runs of each method.

Closely following the MetaSparseINR setup, the network for all tasks is a 4-layer MLP with 256 units, which we meta-train for 2 inner loop steps using a batch size of 3 examples. We use the entire image to compute gradients in the inner loop. In order to correct for the absence of MetaSGD (Li et al., 2017) in MetaSparseINR, we also provide results using a fixed learning rate for SDF & CelebA as a fair comparison, although we strongly suggest to use MetaSGD as a default. Thanks to the kind cooperation of the MetaSparseINR authors, all baseline results are directly taken from their evaluation, ensuring an apples-to-apples comparison.

Full quantitative results across all datasets for varying levels of target sparsity are shown in Figure 3. In all cases we notice a significant improvement, especially at high sparsity levels which are most important for good compression results. At its best, MSCN (with MetaSGD) on CelebA outperforms MetaSparseINR by over 9 decibel at the highest and over 4 PSNR at the lowest sparsity levels considered. As PSNR is calculated on a log-scale this is a significant improvement. To provide a more intuitive sense of those results we provide qualitative results in Figure 4b. Note that at extreme sparsity levels the MetaSparseINR result is barely distinguishable while facial features in MSCN reconstructions are still clearly recognisable. Further qualitative examples are shown in the Appendix.

An interesting aspect of our method is the analysis of resulting sparsity patterns. Figure 4a shows the distribution of sparsity throughout the network at various global sparsity levels. We notice that such patterns vary both based on the overall sparsity level as well as the random initialisation, suggesting that optimal pattern are specific to each optimisation problem and thus cannot be specified in advance to a high degree of certainty. It is also worth pointing out that existing hand-designed sparsity distributions (e.g. Mocanu et al., 2018; Evci et al., 2020) would result in a different pattern, allocating equal sparsity to layers 2-4, whereas our empirical results suggest this might not be optimal in all cases.

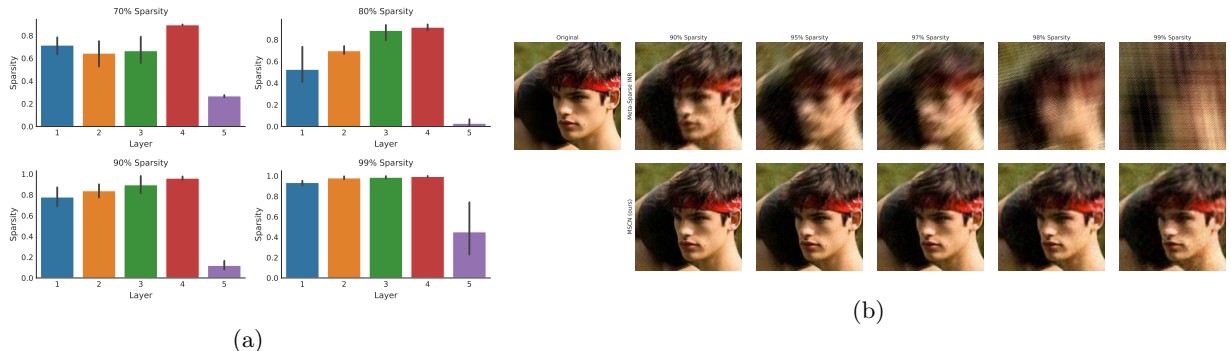

(a)

(b)

Figure 4: Results on CelebA. (a) Sparsity Pattern across layers at various levels of overall network sparsity. Results shown over 3 random seeds. (b) Qualitative results on CelebA for various levels of sparsity.

## 5.2 Structured sparse modulations

| Dataset, array size | Model | Performance at modulation size | | | | |
|---|---|---|---|---|---|---|
| | | 64 | 128 | 256 | 512 | 1024 |
| ERA5, $181 \times 360$ | Functa | 43.2 | 43.7 | 43.8 | 44.0 | 44.1 |
| | MSCN | **44.6** | **45.7** | **46.0** | **46.6** | **46.9** |
| CelebA-HQ, $64 \times 64$ | Functa | 21.6 | 23.5 | 25.6 | 28.0 | 30.7 |
| | MSCN | **21.8** | **23.8** | **25.7** | **28.1** | **30.9** |
| ShapeNet 10, $64^3$ | Functa | 99.30 | 99.40 | 99.44 | 99.50 | 99.55 |
| | MSCN | **99.43** | **99.50** | **99.56** | **99.63** | **99.69** |
| SRN Cars, $128 \times 128$ | Functa | 22.4 | 23.0 | 23.1 | 23.2 | 23.1 |
| | MSCN | **22.8** | **24.0** | **24.3** | **24.5** | **24.8** |

Table 1: Quantitative results for each dataset. Shown is the mean reconstruction error for various modulations sizes. Metrics are Voxel accuracy for ShapeNet 10 and PSNR for all others. Corresponding sparsity levels for MSCN are: 64: 99.1%, 128: 98.2%, 256: 96.4%, 512: 92.9%, 1024: 85.7%.

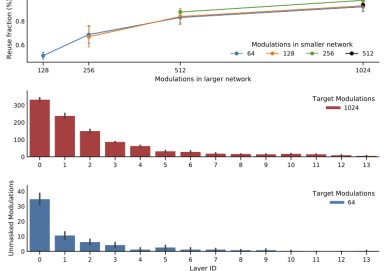

Figure 5: Modulations show a high level of reuse (top). Modulation distribution for 1024 (middle) & 64 modulations (bottom).

In this section, we consider the evaluation of the sparse structured modulation setting (Section 3.3.2). The closest competitor is Functa (Dupont et al., 2022a), which we take to be our baseline method. We evaluate on *Voxels* using the top 10 classes of the ShapeNet dataset (Chang et al., 2015), *NeRF scenes* using SRN Cars (Sitzmann et al., 2019), manifolds using the ERA-5 dataset (Hersbach et al., 2019) and on images using the CelebAHQ dataset (Karras et al., 2017).

Due to the relatively low dimensionality of modulations, networks in this section are significantly deeper, comprising 15 layers of 512 units each. In accordance with Functa, we report results using MetaSGD with 3 inner-loop sets and varying batch sizes (see Appendix for details) due to memory restrictions. We show full quantitative results in Table 1, noting that we outperform Functa by a noticeable margin in almost all settings.

Resulting sparsity patterns shown in Figure 5 (middle & bottom) are particularly interesting, showing that our method leads to the intuitive result of preferring to allocate most of its modulation budget in earlier layers, as such modulations have the potential to have a large effect on the whole network. Indeed Dupont et al. (2022a) write *"reconstructions are more sensitive to earlier modulation layers than later ones. Hence we can reduce the number of shift modulations by only using them for the first few layers of the MLP"*. A possible explanation for that observation is that while early modulations do make the largest difference, our method continues to make use of modulations in later layers. This is another argument for learned sparsity patterns over pre-defined distributions, which would result in a mostly uniform allocation throughout the network.

Interestingly, Figure 5 (top) shows that modulations show a high-degree of reuse. We plot the fraction of modulations that are re-used when starting an optimisation process from the same random initialisation but

allowing for larger number of modulations (x-axis) relative to a network with fewer modulations (distinguished by colours). In all cases the fraction of re-use is much higher than with a random allocation.

Furthermore, we provide qualitative results in Figures 6 and 7. In both cases we observe noticeable improvements over Functa. To provide a qualitative notion of gains afforded by larger number of modulations, we show results for MSCN in Figure 6d.

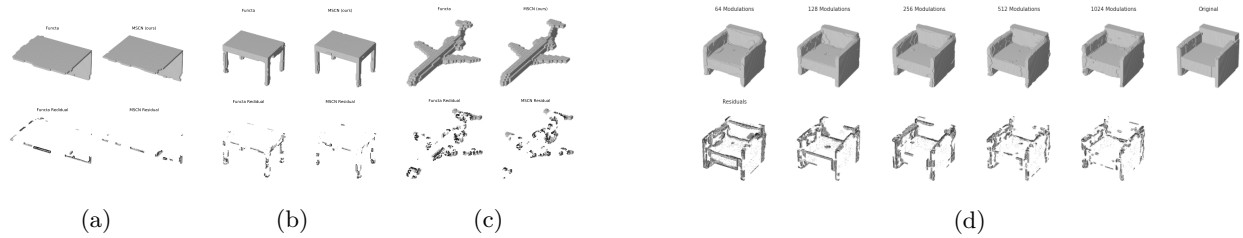

|     |     |     |     |
| :-: | :-: | :-: | :-: |
| (a) | (b) | (c) | (d) |

Figure 6: Quantitative results for structured sparsity on ShapeNet10. (a)-(c) Comparison with Functa for 1024 Modulations (d) Reconstruction quality for increasing number of modulations.

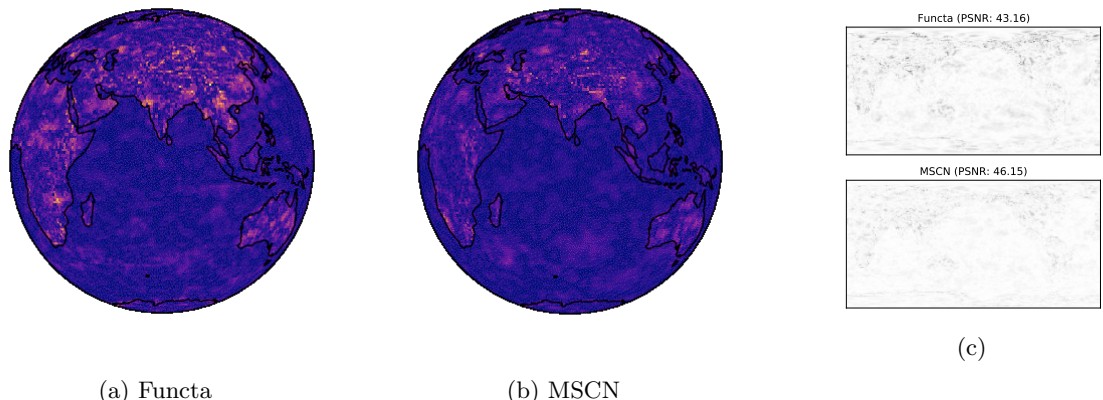

(a) Functa         (b) MSCN

Figure 7: Qualitative results on ERA5 (1024 modulations). (a)-(b) Prediction errors shown on the manifold for 1024 modulation. Best viewed as a .gif (Functa, MSCN). (c) Full error shown over the entire map.

## 5.3 Compression performance

Returning to one of the key motivations of our work, we now provide a comparison with various commonly used compression schemes. The closest method to our work is COIN++ (Dupont et al., 2022b), which is an application of the previously discussed Functa for compression. The authors apply uniform quantisation and entropy-coding to the dense latent vector and compare to a wide range of traditional and learned compression algorithms.

In order for MSCN to be a competitive quantisation scheme, we follow the COIN++ approach to quantisation and entropy coding, which we apply to any non-zero modulations remaining after the trained architecture has has been adapted to a single data item. Identical to the observations for COIN++ we find that modulations can be sparsified to a high level, allowing the quantisation of a standard 32bit representation to only 5-6 bits with little loss of performance. For the large images found in the Kodak dataset, we split a large images into smaller patches that are represented separately. As any shared weights are identical for each example, we consider their cost amortised (i.e. they are part of the compression program) and are thus not reported in the following results. This is standard practice. For this reason, we force the use of identical sparsity patterns for each example (i.e. we do not update $\phi$ in Equation equation 8) to avoid the otherwise necessary cost of communicating the sparsity pattern. Finally, we found the structured sparsity formulation of Section 3.3.2 to

be most suitable for optimal compression, combining inductive biases with structure learning algorithms. We provide further details on these experiments in the Appendix.

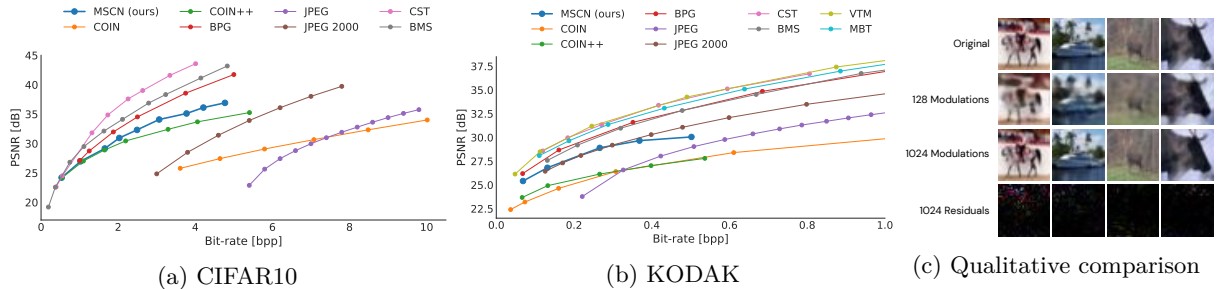

(a) CIFAR10       (b) KODAK       (c) Qualitative comparison

Figure 8: Rate-distortion plots on CIFAR10 (a) and Kodak (b). (c) shows a qualitative comparison on CIFAR10. The 128/1024 modulation rows correspond to evaluating to the leftmost and rightmost point of the MSCN curve in (a). Results are almost indistinguishable for 1024 modulations. Best viewed on a computer.

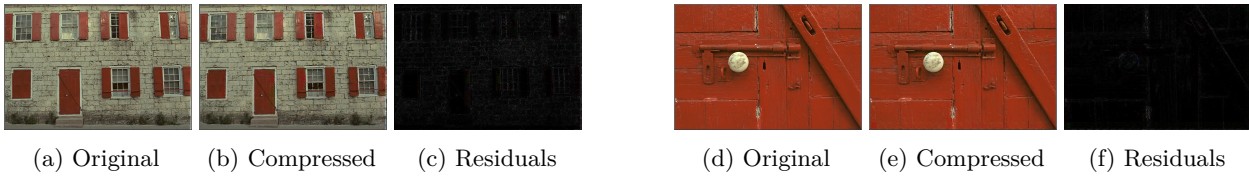

(a) Original    (b) Compressed    (c) Residuals        (d) Original    (e) Compressed    (f) Residuals

Figure 9: Qualitative examples on the Kodak dataset. The PSNRS achieved are 25.58 (left) and 31.77 (right).

We provide a comparison with various compression schemes for CIFAR10 in Figure 8a and Kodak (for which we pre-train on the DIV2K dataset (Agustsson & Timofte, 2017) as in (Strümpler et al., 2021)) in Figure 8b. The x-axis shows the bit-rate in bits-per-pixel [1]. We note that MSCN consistently outperforms COIN++, suggesting that the previously observed performance improvements over Functa also hold for compression applications. We also find that our method is competitive specifically in the low bit-rate regime, where we achieve strong performance improvements over JPEG/JPEG2000. Note that the strong improvement over COIN++ on Kodak may be a result on the pre-training on Div2k (COIN++ uses frames of Vimeo90k), whereas the results on CIFAR10 are in line with what can be expected given the improvement over Functa in the previous section. Qualitative results are shown in Figures 8c and 9.

As hypothesised in COIN++ the resulting gap to state-of-the-art codecs like BMS & CST may be due to their use of deep generative models for entropy coding which could be added to our formulation in future work with little conceptual work. Moreover, the use of more intelligent post-training compression has recently proven fruitful (Strümpler et al., 2021) and should further improve results. It is however important to state that such methods require significantly higher encoding times, thus being in conflict with one of our key motivations, We thus avoid a direct comparison as this would likely fail to communicate the inherent trade-off of quality versus compression speed. As with deep entropy coding, such techniques could be straight-forwardly used in conjunction with MSCN, demonstrating the flexibility of our method.

### 5.4 Discussion & Future work

In this work we have introduced a principled framework for sparse Meta-Learning, demonstrating two instantiations particularly suitable for Implicit Neural Representations. Our extensive evaluation show competitive results, outperforming various state-of-the-art techniques. It is worth mentioning that the ideas introduced in this work can be straight-forwardly combined with some of the considered baselines. In the Functa case for instance, it would be reasonable to expect a latent variable approach to be more competitive if only a sparse subset of modulations needs to be reconstructed.

---

[1]$\text{BPP} = \frac{\text{Bits per param.} \times \text{Number of param.}}{\text{Data dim.}}$

A key motivation was the goal of avoiding costly architecture search procedures used in related work (e.g. Dupont et al., 2021; Strümpler et al., 2021). We have shown that in both cases of structured and unstructured sparsity, we observe sparsity patterns that differ from previously hand-designed distributions and also adopt to the specific initialisation of the weights. In the structured sparsity case, we observe that this can be combined with inductive biases which have previously been found to useful.

Our method continues the trend of rapid advances made with INRs for compression. As the field continues to challenge the state-of-art in compression, we observe that sparsity is likely to be a key element in this endeavour. We further hypothesise that additional improvements are likely to come from alternative Meta-Learning techniques which avoid the high memory requirements of MAML. In addition, we expect methods with smarter quantisation and based on deep entropy coding to significantly improve results over our simple baseline. Current progress nevertheless inspires optimism for the prospect of a single learning algorithm that can be applied as a compression method to a vast set up modalities.

In the wider context of Meta-Learning, we anticipate this framework to be particularly suitable in applications where fast inference is required. Sharing the gates introduced in Section 3 among sets of parameters is an easy way of introduced group sparsity and thus reducing floating point operations required in the forward pass. A popular example would be on-device recommender systems (Galashov et al., 2019)). Furthermore, the introduced framework could be used in Continual Meta Learning as previously done in (e.g. Mallya & Lazebnik, 2018; Von Oswald et al., 2021; Schwarz et al., 2021).

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
