# OpenReview forum: "Meta-Learning Sparse Compression Networks"
_TMLR — Accepted by TMLR_

### Review · Reviewer_tps1 · 2022-06-15

**Summary Of Contributions:**

The paper introduces a method for data compression based on the idea of encoding the data in the weights of a neural network which maps coordinates to signal values. This has been explored before, but the present work uses a better method for sparsification of the weights based on a differentiable L0 penalty, and shows how this can be combined with a meta learning approach that significantly improves encoding time as well as compression rates. The method is evaluated on a range of datasets with quite different kinds of data such as images, data on manifolds and shapes.





**Requested Changes:**

- Report standard metrics on standard datasets to enable direct comparison to existing methods based on INRs and other methods, as discussed above.
- Discuss difference to Strumpler and other related work if deemed relevant
- Eq. 3, missed a min and a subscript 0, I think


**Strengths And Weaknesses:**

The idea of this paper makes sense to me, and the project is well executed. The paper is clearly written, and explains the important ideas well. Experimental validation is fairly extensive.

The main weakness is that due to the way results are presented, it is not clear how this method compares to state of the art (neural) compression methods. The main issues are that the datasets are non-standard in the compression literature, and the results are not presented as R/D curves (bits per pixel vs PSNR) but rather as "Fraction of surviving parameters" vs PSNR. I'm sure the later can be turned into BPP, but it would be nice to not leave this job to the reader. Also, the result would be most convincing if the BPP is actually computed from the filesize on disk rather than a theoretical calculation (many practical issues can come up when trying to actually encode the data). Another good metric to report is BD-rate.

Regarding the dataset, it would be nice if the authors could run their method on one of the standard benchmarks, such as KODAK which is quite small.

Even if the method is not competitive with the state of the art (non-INR based methods), I would likely recommend acceptance because the method is interesting and promising (perhaps with further innovations), and method is very flexible, likely yielding at least decent results on a wide range of data modalities, with fairly minimal effort.

Regarding novelty, I am not completely up to date with the literature, but it seems like Strumpler et al. also already considered sparsification, which is not mentioned although Strumpler is cited. They use an L1 penalty instead of L0. Comparing to this method would be nice. Other papers based on optimizing parameters for compression include Yang et al. and van Rozendaal et al.


Strumpler, Postels, Yang, Van Gool, Tombari, Implicit Neural Representations for Image Compression
Yang, Bamler, Mandt, Improving inference for neural image compression
van Rozendaal, Huijben, Cohen, Overfitting for Fun and Profit: Instance-Adaptive Data Compression

---

> ### Author Response · Authors · 2022-06-27
> **Respose to reviewer tps1**
>
> We thank the reviewer for their comments and generally agree with all of the points raised. In particular:
>
> 1) The reviewer’s comment is in line with feedback from other reviewers and makes it clear that the compression experiments/rate-distortion plots (Figure 7) deserve additional attention and in particular a comparison to (Dupont et al., 2022b) and/or (Strümpler et al., 2021) on a standard dataset (Kodak or the standard version of CelebA) to make a clear case for this as a compression method. This will include the use of default techniques such as quantisation and entropy coding which can be straight-forwardly used for any weights that have not been sparsified. We will deliver those experiments as soon as possible but would very kindly ask the reviewer to hold off from submitting their finalised review and decision recommendation (Deadline: Jul 14) until we can deliver those results. While the authors make those experiments their main priority any leeway here would be much appreciated.
> 2) We do indeed report the BPP as a theoretical quantity (see Response to reviewer R1Eu for details on how it is computed) but agree that the actual size on disk should also be reported. As mentioned above, this will also make it clear how this will interact with standard tricks such as post-training quantisation and entropy coding. We will include those results.
> 3) Regarding the use of "Fraction of surviving parameters" instead of BPP/PSNR: This is a valuable comment. The reason why we also report sparsity levels is primarily to be able to directly compare to MetaSparseINR without access to their trained models, but also because such metrics are very common in the sparsity literature and would likely be requested if missing. Finally, it is also worth noting that the Sparse Meta-Learning algorithm we present is more general and might be used for other use-cased beyond compression, in which cases the sparsity metric may be more appropriate.
> 4) Regarding the use of L1 regularisation it should be noted that Strumpler et al. use the term “sparsity” in a somewhat non-standard way and argue “we have the same goal of limiting the entropy of the weights” which makes sense due their heavy use of quantisation. The goal in NN sparsity research would however not be to limit the entropy of non-zero weights, as this could reduce expressiveness in an already heavily constrained setting. Indeed, modern NN sparsity research does not typically impose shrinkage on the non-zero parameter values themselves (e.g. Jayakumar et al., 2020, Utku et al. 2020). It is also indicative that L1 regularisation is not typically used as a comparison in modern Sparse NN research as it is likely to lead to significantly worse results since it does not encourage parameters to become exactly zero. We accept however, that in cases where non-zero modulations are further entropy-coded this loss could be useful in addition to an L0 objective (Louizos at el., 2017 also make this point and explicitly explain how such regularisation terms can be used in conjunction if desired). We will strive to provide such a comparison in addition to the promised experiments in 1).
>
> Louizos, Christos, Max Welling, and Diederik P. Kingma. "Learning sparse neural networks through $ L_0 $ regularization." arXiv preprint arXiv:1712.01312 (2017).
>
> Jayakumar, Siddhant, et al. "Top-kast: Top-k always sparse training." Advances in Neural Information Processing Systems 33 (2020): 20744-20754.
>
> Evci, Utku, et al. "Rigging the lottery: Making all tickets winners." International Conference on Machine Learning. PMLR, 2020.

---

> ### Author Response · Authors · 2022-07-10
> **Revision**
>
> Given the proximity to the review deadline we would like to let the reviewer know that we have uploaded a revision of our work with the most important feedback in the review process so far: The requested comparison to various compression schemes (Figure 8) on Cifar10 and Kodak. While both COIN++ (Dupont et al., 2022) and the technique introduced in (Strumpler et al., 2021) were suggested, we found that COIN++ is a more realistic comparison to our technique and thus was the basis of our attempt to improve our work in this regard. The main reason for this decision is that (Strumpler et al., 2021) propose multiple quantisation schemes at inference time that require re-training the network in multiple rounds, thus being in conflict with our stated goal of allowing fast encoding at inference time. In addition, the authors make heavy use of architecture search (see section 4.4) which we also expressly argued to avoid in this paper. For this experiment we thus follow COIN++ and make use of the proposed quantisation and entropy coding which is extremely fast and easy to compute. In our case we directly apply this quantisation to any non-zero modulations.
>
> We find that our work compares favourably with COIN++, outperforming the method in both setups. We discuss various strategies for addressing the remaining gap in the revised manuscript.
>
> In the next couple of days leading up to the final review deadline, we will continue to address the reviewer's questions through updates to the manuscript (also qualitative examples for Kodak/Cifar10) but wanted to share this larger update ahead of time to allow more flexibility to the reviewer. Thank you!

---

### Review · Reviewer_R1Eu · 2022-06-15

**Summary Of Contributions:**

The main topic of this paper is lossy compression (LC) using implicit neural representations (INRs). In previous works on LC with INRs, encoding a message (e.g. image) is done by overfitting a neural network to learn a function from the spatial coordinates of images to the RGB values. The weights of the network are then stored and represent the code for that specific message. To decode, the network weights are restored and the function is evaluated to recover the message.

The author's contributions are two-fold:
1. (Reducing the required rate) The size of the network defines the final code-length. Therefore the authors incorporate state-of-the-art (according to them) learned-sparsity techniques to reduce the number of weights which should bring down code-length. This technique is backpropagation-compatible, which is crucial for point (2.) below.

2. (Reducing the required compute) Instead of overfitting a network to each message, a meta-learning approach (MAML) is used to learn a common initialization across a set of messages.



**Broader Impact Concerns:**

No broader impact statement was available. The authors may want to consider discussing known algorithmic biases for model pruning, and how that may disproportionately affect underrepresented features in the data when working with images of humans. See https://arxiv.org/abs/2010.03058

**Requested Changes:**

1. (critical) I believe the authors should clarify how the experiments were conducted that resulted in figure 7. See previous section of this review.

All below are optional, but I'd highly recommend the following to strengthen the paper:

2. Usually other distortion measures, such as MS-SSIM, are placed alongside PSNR as they capture perceptual quality better (empirically).

3. This paper might be of interest and can serve as a motivation for the introduction: https://proceedings.mlr.press/v151/isik22a.html. They show that any good model compressor, under certain conditions, must perform some sort of pruning/sparsity.

4. Maybe the author could comment and give a sense of how much less time/steps/resources were required in the experiments, as mentioned before.

5. Writing suggestions. Nothing major, but might want to consider.
- I found the discussion of part (ii) in section 3.2 a bit confusing. I assume it refers to using $\theta + \delta\theta\circ\mathbf{z}’$ as discussed in the third paragraph on page 5. If so, I’d recommend merging that later paragraph into part (ii) to make it clearer.
- I believe equation (6) has a typo. The first 2 summations have the same indexing variable $i$.
- What does “highest plasticity” mean on page 5? Maybe make that clearer (unless it is a widely used term that I’m not aware of).
- Typo on page 6: “for ease of ~notion~ notation”
- Typo on page 9: “modulations in ~layer~ later layers”
- Vector notation is inconsistent. Some vectors are bold such as $\mathbf{x}$ and $\mathbf{y}$, but $\theta, \phi, s$ are not.


**Strengths And Weaknesses:**

I don’t know much about meta-learning, so I’ll comment on the aspects regarding compression.

Overall the paper looks technically correct and I don’t see any major technical issues with the method. However, the experiments and discussions for lossy compression are a bit lacking. The exact setup of the experiments that resulted in Figure 7 need to be clarified.

**Note that it is quite possible that I did not completely understand how MAML works or is used.**

In detail:

1. How exactly would a compression scheme with this method work? You are given a sequence of images $X_1, \dots, X_n$ to compress. MAML is used to learn $(\theta_0 + \delta\theta_i)\circ\mathbf{z}^\prime$ for $i=1,\dots,n$ images. Then, what exactly is used to calculate the rate in Figure 7? Is it just the parameters $\\{\delta\theta_i\circ\mathbf{z}^\prime\\}_{i=1}^n$ or does $\theta_0\circ\mathbf{z}^\prime$ also enter the calculation? I'm assuming both are accounted for in the rate, as that would be the correct calculation, but the authors should clarify this.

2. Is the method highly sensitive to the number of images being compressed (i.e. $n$ above)? As you compress more images, $\theta_0$ needs to be a good initialization point for more and more samples. If so, it would only make sense to talk about the rate-distortion curve (Figure 7) as a function of $n$, especially because JPEG/JPEG2000 won't have a reduction in performance as $n$ increases (assuming it was applied individually to each image).

4. The idea of MAML was to reduce the overall compute necessary to compress a set of images with respect to previous methods using INRs. Maybe the author could comment and give a sense of how much less time/steps/resources were required in the experiments.

5. The wording of the introduction, abstract and final paragraph of section 5.2 suggest that the main objective is compression. The paper mostly focuses on evaluating the resulting sparseness of the method. This makes sense because the weights are not entropy coded and are instead stored to disk using 16 or 32bit precision. However, this is mentioned only at the very end of page 9 which kept me wondering while reading the paper why the authors focused on the sparsity and not the actual rate (e.g. BPP). Maybe this could be mentioned at the start of the paper.

---

> ### Author Response · Authors · 2022-06-27
> **Response to reviewer R1Eu**
>
> We thank the author for their detailed feedback, helpful comments and particular attention to details.
>
> Strengths and weaknesses section:
>
> 1) It is worth mentioning the distinction between training and inference/test time here. $\textbf{Training time}$: The result of the training stage will be an initialisation of weights $\theta_0$ and mask parameters $\phi_0$ that has been obtained through the MAML procedure. This initialisation is shared among all images $X^{(1)},\dots,X^{(n)}$ that are to be compressed at inference time. We can thus assume its compression cost is amortised and can be neglected (this is standard in the literature). $\textbf{Inference time}$: When a particular image $X^{(i)}$ is to be compressed, we perform a very small, fixed number of optimisation iterations (e.g. 3 for the experiments in Section 5.2) to arrive at specialised weights $\theta’^{(i)}$. We can then compute the difference between the known initialisation $\theta_0$ and the specific parameter changes used to model $X^{(i)}$: $\delta\theta^{(i)} = \theta’^{(i)} - \theta_0$. Now, $\delta\theta^{(i)}$ will zero by design at indicies $j$ whenever $\mathbf{z}’^{(i)}_j$ = 0. And thus we only need to store the values in $\theta’^{(i)}$ which are non-zero. The compression cost is thus the number of non-zero entries in $\mathbf{z}’^{(i)}$ times the compression cost of each entry in $\delta\theta^{(i)}$ (e.g. 32 bits for float32). A single point in Figure 7 is thus computed as the average across all data points in the test set. The rate distortion plot is obtained by varying the sparsity of $\mathbf{z}’^{(i)}$ through the $L_0$ penalty hyperparameter $\lambda$ (eq. 8). Does this make sense? (Note that $\mathbf{z}’^{(i)}$ has been computed from the updated $\phi’^{(i)}$ but this does not need to be stored.)
>
> 2) As mentioned above, the compression cost at inference time is specific to each individual image and thus does not depend on the number of examples being compressed (thus Figure 7 does not depend on $n$). The quality of the MAML initisalition $\theta_0$ on the other hand does of course depend on the number of datapoints available in the training set, but related work (e.g. Strümpler et al., 2021) shows that even a relatively small training set of about 1000 images is sufficient for such methods to compete with hand-designed compression methods. In the age of the internet we argue that it is reasonable to assume that a large enough training set could be collected, even for more complex data types. The diversity of our experiments shows this is indeed the case for many problems.
>
> 3) As mentioned in 1), the key metric is the compute cost used at inference time, and this is only about 2-3 steps of gradient descent, which is very cheap in comparison to normal NN training. Training cost on the other hand is on the order of the cost used to train Functa, which we will provide further detail for. It is fair to say that an advantage of hand-designed methods such as JPEG/JPEG 2000 is their absence of any training cost. However, this is true for all learned compression methods and not particular to this submission.
>
> 4) This is a valuable comment. The reason why we also report sparsity levels is primarily to be able to directly compare to MetaSparseINR without access to their trained models, but also because such plots are very common in the sparsity literature and would likely be requested if missing. Finally, it is also worth noting that the Sparse Meta-Learning algorithm we present is more general and might be used for other use-cased beyond compression, in which cases the sparsity metric may be more appropriate.
>
> Requested changes section:
>
> 1) The reviewer’s comment is in line with feedback from other reviewers and makes it clear that the compression experiments/rate-distortion plots (Figure 7) deserve additional attention and in particular a comparison to (Dupont et al., 2022b) and/or (Strümpler et al., 2021) on a standard dataset to make a clear case for this as a compression method. This will include the use of default techniques such as quantisation and entropy coding which can be straight-forwardly used for any weights that have not been sparsified. We will deliver those experiments as soon as possible but would very kindly ask the reviewer to hold off from submitting their finalised review and decision recommendation (Deadline: Jul 14) until we can deliver those results. While the authors make those experiments their main priority any leeway here would be much appreciated.
>
> 2) & 3) & Broader Impact Statement: Thank you for those useful suggestions. We will try to include this in the manuscript.
>
> 5) Thank you for the very detailed feedback. Much appreciated!

---

> > ### Comment · Reviewer_R1Eu · 2022-06-28
> > **Followup**
> >
> > Thanks for the clarifications.
> >
> > Have the authors considered using previous samples to update the weights? At step $i$ the sequence of previous samples, $X^{(1)}, \dots, X^{(i-1)}$, is available at the decoder, so this could be used to improve the rate. A more precise example would be to, say, keep a running update of $\theta_0$; i.e. something like $(X^{(i)}, \theta_{i-1}) \mapsto \theta_i$. Maybe by running an extra step of MAML (or something less compute intensive) as symbols are encoded/decoded.
> >
> > > We will deliver those experiments as soon as possible but would very kindly ask the reviewer to hold off from submitting their finalised review and decision recommendation (Deadline: Jul 14) until we can deliver those results
> >
> > OK on my side.

---

> > > ### Author Response · Authors · 2022-06-29
> > > **Response to Followup**
> > >
> > > Perhaps the closest idea to the proposal that the authors are aware of would be to keep an exponential moving average of the weights, i.e. $\widetilde{\theta_t} = \alpha \theta_t + (1-\alpha) \widetilde{\theta}_{t-1}$  (where $\widetilde{\theta}_0 = \theta_0$ and $t$ indicates the $t$-th update step) and evaluate with $\widetilde{\theta}_T$ instead of $\theta_T$. This is indeed a trick that is sometimes used in Meta-Learning and does occasionally improve performance (depending on the specific optimisation landscape). We have not introduces this trick for MSCN yet as a fair comparison would also involve re-running Functa with this idea. We could however include this as an additional ablation study, similar to how we run MSCN without MetaSGD in Figure 2.
> > >
> > > Thank you for the suggestion!
> > >
> > > Update: We've included this idea as an ablation study in the Appendix. It does improve things slightly for some models and never hurts. We think this might be connected to the sharpness of the loss landscape around the MAML initialisation, although a thorough analysis of this is probably outside the scope of this work :)

---

> ### Author Response · Authors · 2022-07-10
> **Revision**
>
> Given the proximity to the review deadline we would like to let the reviewer know that we have uploaded a revision of our work with the most important feedback in the review process so far: The requested comparison to various compression schemes (Figure 8) on Cifar10 and Kodak. While both COIN++ (Dupont et al., 2022) and the technique introduced in (Strumpler et al., 2021) were suggested, we found that COIN++ is a more realistic comparison to our technique and thus was the basis of our attempt to improve our work in this regard. The main reason for this decision is that (Strumpler et al., 2021) propose multiple quantisation schemes at inference time that require re-training the network in multiple rounds, thus being in conflict with our stated goal of allowing fast encoding at inference time. In addition, the authors make heavy use of architecture search (see section 4.4) which we also expressly argued to avoid in this paper. For this experiment we thus follow COIN++ and make use of the proposed quantisation and entropy coding which is extremely fast and easy to compute. In our case we directly apply this quantisation to any non-zero modulations.
>
> We find that our work compares favourably with COIN++, outperforming the method in both setups. We discuss various strategies for addressing the remaining gap in the revised manuscript.
>
> In the next couple of days leading up to the final review deadline, we will continue to address the reviewer's questions through updates to the manuscript (also qualitative examples for Kodak/Cifar10) but wanted to share this larger update ahead of time to allow more flexibility to the reviewer. Thank you!

---

> ### Author Response · Authors · 2022-07-14
> **Resource usage**
>
> We've added a paragraph in the Appendix to try and answer the resource usage question. In short, our method will have a faster runtime than the two methods mentioned in the discussion (Strümpler et al., 2021; Dupont et al., 2022b) based on the method design - COIN++ needing an additional latent->modulation network, Strümpler et al. using expensive quantisation aware training stages at inference time.
>
> In terms of the resources used for training we've done to match the number of training steps and batch size and always evaluated for the same number of inner loop steps than all baseline results. We provide a detailed list of all hyperparameters in the Appendix but have made genuine effort for the experiments to be as closely comparable and fair as possible.

---

### Review · Reviewer_B3Hu · 2022-06-16

**Summary Of Contributions:**

The manuscript proposes a new method for sparsifying parameter updates in meta-learning, specifically within the context of data compression with INRs (implicit neural representations). The method is based on imposing an additional L_0 penalty (Louizos et al., 2017) on the model update ($\delta \theta$) in the MAML framework. Following Louizos et al., 2017, the Gumbel-Softmax trick is used to differentiate through samples of the binary distributions masking the weights, so the modified MAML inner loop is still differentiable (for the purpose of computing the gradient of the overall meta-learning objective), allowing for end-to-end (meta) training.
The proposed method is evaluated on several datasets, demonstrating superior sparsity v.s. reconstruction quality tradeoff, especially compared to a closely related method (Meta-SparseINR; Lee et al., 2021) which performs iterative pruning in the MAML outer loop. However, I have concerns about how meaningful the experimental comparisons are.

**Broader Impact Concerns:**

There are no obvious ethical implications of this work that need to be addressed in the manuscript.

**Requested Changes:**

1. As per weakness #1, adding results on a standard image compression benchmark in terms of bitrate v.s. PSNR,  against INR compression baselines such as (Strümpler et al., 2021; Dupont et al., 2022b), would make a much more convincing case for the viability of the proposed method for compression.

2. As per weakness #2, more details for the evaluation setup regarding where the sparsity numbers come from ($\delta \theta$ v.s. something else) for each method would add significant insight into the comparisons and behavior of the different methods.

3. Minor typo on page 4,  right above Eq(8): "the other loop" -> "the outer loop".



**Strengths And Weaknesses:**

Strengths:

1. The method has broad applicability, not only to INR, but also other meta-learning tasks where a sparse model (or model update) is required -- although further experiments may be needed to verify this.

2. The method is conceptually straightforward, and allows end-to-end training and straightforward implementation with standard tools from deep learning.


Weaknesses:

1. Lack of comparison with baselines that directly targets the bit-rate cost of compression with INR.
As the author write early on in the manuscript, "we specifically focus on improving the suitability of INRs as a compression method", and therefore argue (correctly) that we should focus on the compression cost of the parameter difference $\delta \theta$ that is transmitted during data compression, instead of the updated model $\theta_0 + \delta \theta$.  Now, the sparsity of $\delta \theta$ is only a surrogate for its compressibility, and at the end of the day compression algorithms are evaluated on the file size (bitrate). The lack of such comparison makes it hard to evaluate the success/failure of the sparsity-based method towards the stated goal of improving data compression with INR.

2. It's unclear what is being compared across some of the methods --- the sparsity of $\delta \theta$, or $\theta$ ($\theta_0 + \delta \theta$) itself. Clearly, the proposed method targets the sparsity of $\delta \theta$ -- "Note that as $\theta_0$ is dense, the resulting
$\theta_0 + \delta \theta$ is still dense". By contrast, the Meta-SparseINR (Lee et al., 2021) baseline aims to sparsity $\theta_0$, and subsequently $\theta_0 + \delta \theta$. It's not surprising that, with a denser $\theta_0 + \delta \theta$  (although still sparse $\delta \theta$), the proposed MSCN method outperforms Meta-SparseINR in reconstruction quality. It would be rather meaningless if the comparison is between the sparsity level of $\delta \theta$ for the proposed MSCN method, v.s. the sparsity level of $\theta_0 + \delta \theta$ for the Meta-SparseINR baseline.

---

> ### Author Response · Authors · 2022-06-27
> **Response to reviewer B3Hu**
>
> We thank the author for their feedback and helpful comments.
>
> 1) In line with feedback from other reviewers it is clear that the compression experiments/rate-distortion plots (Figure 7) deserve additional attention and in particular a comparison to (Dupont et al., 2022b) and/or (Strümpler et al., 2021) on a standard dataset to make a clear case for this as a compression method. This will include the use of default techniques such as quantisation and entropy coding which can be straight-forwardly used for any weights that have not been sparsified. We will deliver those experiments as soon as possible but would very kindly ask the reviewer to hold off from submitting their finalised review and decision recommendation (Deadline: Jul 14) until we can deliver those results. While the authors make those experiments their main priority any leeway here would be much appreciated.
>
> 2) We fully agree with the reviewer that a direct comparison between MetaSparseINR and a sparsification of $\theta + \delta\theta$ with our method would constitute the most direct apples-to-apples comparison despite not being the most natural approach to optimal compression. The reviewer is correct that the current comparison is between the sparsity of $(\theta + \delta\theta)$ for MetaSparseINR but only $\delta\theta$ for our method. The reasoning behind this was that the direct sparsification of only $\delta\theta$ is not actually possible for MetaSparseINR (due to non-differentiability of their pruning procedure) despite being the preferred means for compression. However, as this is straightforward to implement in our framework, we will endeavor to provide those results as soon as possible.
>
> Finally, we will fix the typo mentioned. Thank you for the attention to detail!

---

> > ### Comment · Reviewer_B3Hu · 2022-07-04
> > **Thank you for your response**
> >
> > The proposal looks good to me, and thanks for the clarifications esp. regarding figure 7.
> >
> > BTW, I found a new typo: you may want to change $\min_\theta$ to $\arg \min_\theta$ in Eq (3) and similarly other equations.

---

> ### Author Response · Authors · 2022-07-10
> **Revision**
>
> Given the proximity to the review deadline we would like to let the reviewer know that we have uploaded a revision of our work with the most important feedback in the review process so far: The requested comparison to various compression schemes (Figure 8) on Cifar10 and Kodak. While both COIN++ (Dupont et al., 2022) and the technique introduced in (Strumpler et al., 2021) were suggested, we found that COIN++ is a more realistic comparison to our technique and thus was the basis of our attempt to improve our work in this regard. The main reason for this decision is that (Strumpler et al., 2021) propose multiple quantisation schemes at inference time that require re-training the network in multiple rounds, thus being in conflict with our stated goal of allowing fast encoding at inference time. In addition, the authors make heavy use of architecture search (see section 4.4) which we also expressly argued to avoid in this paper. For this experiment we thus follow COIN++ and make use of the proposed quantisation and entropy coding which is extremely fast and easy to compute. In our case we directly apply this quantisation to any non-zero modulations.
>
> We find that our work compares favourably with COIN++, outperforming the method in both setups. We discuss various strategies for addressing the remaining gap in the revised manuscript.
>
> In the next couple of days leading up to the final review deadline, we will continue to address the reviewer's questions through updates to the manuscript (also qualitative examples for Kodak/Cifar10) but wanted to share this larger update ahead of time to allow more flexibility to the reviewer. Thank you!

---

> ### Author Response · Authors · 2022-07-12
> **Weakness 2 & Requested Change 2**
>
> As promised we have added an ablation study that compares the originally proposed sparsification $\mathbf{z}\odot\delta\theta$ using MSCN with a sparsification of $\mathbf{z}\odot(\theta_0 + \delta\theta$). This can be found in the supplementary material (Section C: Performance for a fully sparse network) or for your convenience here: https://drive.google.com/file/d/1R2kCs8RRBwVLOgSXpB8ILJHiD4eIzAc5/view?usp=sharing
>
> In short, we find that our framework is still competitive when a fully sparse network is desired, outperforming MetaSparseINR. Nevertheless, the advantage of our formulation is clear, in that it allows us to gain a further improvement through the more intuitive sparsification of $\delta\theta$ only.

---

### Comment · Action_Editors · 2022-06-24
**Starting discussion period**

Dear Authors,

All reviews have been posted and it is time to engage in a discussion with the reviewers. Some of the reviewer concerns are summarized below, but the reviews contain more detail:

- Lack of baselines that directly target the bit-rate cost (the current focus is on sparsity which is only a proxy)
- Confusions around which parameters are counted towards the bit rate and which sparsity is compared across methods
- Questions about runtime and other resource usage
- Non-standard benchmark data sets

I look forward to the discussion.

Best,
Your Action Editor

---

> ### Author Response · Authors · 2022-07-14
> **End of discussion period**
>
> Dear action editor,
>
> We've done our best to actively engage in the discussion and provide additional results or comments for all the most important points raised during the discussion period. Specifically:
>
> - Confusions around which parameters are counted towards the bit rate and which sparsity is compared across methods: We have done our best to explain the compression pipeline, adding a model diagram in the main text and a detailed explanation in the Appendix
> - Confusions around which parameters are counted towards the bit rate and which sparsity is compared across methods
> - Lack of baselines that directly target the bit-rate cost (the current focus is on sparsity which is only a proxy) / Non-standard benchmark data sets: We have addressed this by providing results including a plethora of methods on CIFAR10 & Kodak
> - Questions about runtime and other resource usage: We have addressed this in the Appendix
>
> In case any further comments are raised in the reviewer's final feedback, we'd do our best to include this in a camera-ready version. We appreciate the feedback by all authors and hope this will help you in making an appropriate decision.

---

### Decision · Action_Editors · 2022-08-02

**Recommendation:** Accept as is

**Comment:**

All reviewers agreed that the paper was innovative and technically correct. The authors have successfully addressed the reviewers' concerns on clarity, novelty, metrics, and baseline comparisons during the review period. These extensive improvements helped make the approach more comparable to earlier work in the field. No further revisions are necessary.